# Detection of Relevant Heavy Metal Concentrations in Human Placental Tissue: Relationship between the Concentrations of Hg, As, Pb and Cd and the Diet of the Pregnant Woman

**DOI:** 10.3390/ijerph192214731

**Published:** 2022-11-09

**Authors:** Soledad Molina-Mesa, Juan Pedro Martínez-Cendán, Daniel Moyano-Rubiales, Inmaculada Cubillas-Rodríguez, Jorge Molina-García, Ernesto González-Mesa

**Affiliations:** 1Doctoral School of San Antonio Catholic, University of Murcia Guadalupe de Maciascoque, 30107 Murcia, Spain; 2Biomedical Research Institute of Malaga (IBIMA), Research Group in Maternal-Fetal Medicine, Epigenetics, Women’s Diseases and Reproductive Health, 29071 Malaga, Spain; 3Department of Obstetrics and Gynecology, San Antonio Catholic University of Murcia, Guadalupe de Maciascoque, 30107 Murcia, Spain; 4Obstetrics and Gynecology Service, Regional University Hospital of Malaga, 29011 Malaga, Spain; 5Surgical Specialties, Biochemistry and Immunology Department, Málaga University, 29071 Malaga, Spain

**Keywords:** heavy metals, placenta, cadmium, arsenic, lead, mercury, contaminants, pregnancy, toxicity, dietary habits

## Abstract

Heavy metals can cross the placental barrier and reach the fetal compartment, threatening fetal development. Pregnant women can acquire these through food, drinking water, toxic habits or simply by breathing polluted air. The placenta has been described as a biomarker of maternal and fetal exposure to different toxic elements. Objectives: The main objective of this study was to test the possible existence of heavy metal deposits (Pb, As, Cd and Hg) in the placentas of women who gave birth at term in our setting, analyzing the influence of daily life and dietary habits. Methods: We studied 103 placentas, obtained by consecutive sampling, of women that delivered in the Regional Maternity Hospital of Malaga between March and June, 2021. As, Cd and Pb concentrations were analyzed using mass spectrometry techniques. Hg concentration was studied according to US EPA method 7473. Women also answered a questionnaire with epidemiological variables. Results: Detectable concentrations were found in 14.56% [As], 44.6% [Cd], 81.5% [Pb] and 100% [Hg]. [Pb] and [As] correlated significantly (Spearman’s Rho of 0.91 and <0.001), as did [Hg] and [Cd] (Spearman’s Rho 0.256, *p* < 0.004). The [Pb] and [AS] concentrations were significantly higher in cases of tap water consumption. [Hg] concentrations predicted the birth weight of female newborns.

## 1. Introduction

Prenatal life is very susceptible to possible contaminants or teratogens that may compromise pregnancy and the future health of the child. Exposure to heavy metals during organogenesis can cause permanent changes in structure and anatomy, while if it occurs in the final stages of pregnancy, it can cause functional alterations [1]. The immune, respiratory and central nervous systems are also vulnerable to postnatal exposure as they are not fully developed at birth and need to undergo an extended period of postnatal maturation. The human fetus is completely dependent on the placenta, which acts as a filter reducing the passage of harmful substances and protecting it. However, it has been shown that the placenta cannot prevent the passage of certain teratogens. The placenta is, therefore, the interface between the mother and the outside world and the developing fetus, anchoring the developing embryo to the uterine wall and connecting it to the maternal bloodstream [2]; thus, the use of placental tissue has been proposed as a biological matrix of exposure for different organic and inorganic contaminants [3,4]. In the study of these pollutants, it is of great interest to know the level of exposure to heavy metals such as mercury, lead, arsenic and cadmium since they can have harmful effects on fetal and postnatal child development. These pollutants can be found in our daily lives in water, contaminated food, tobacco smoke, fuels, products released by burning coal, fertilizers, etc. These heavy metals are a threat to fetal well-being, as they cross the placenta and accumulate in fetal tissues. Prenatal exposure to mercury and lead poses a risk to babies’ developing brains, while exposure to lead and cadmium has been correlated with reduced birth weight and size [5].

Traditionally, the placenta has been considered as a barrier that prevents the passage of harmful substances protecting the fetus from toxic exposures. Placental cells express certain proteins to carry out the transport of unwanted substances back into the maternal circulation, as well as to participate in the retention and detoxification of toxic substances [6]. However, it has been proven that the placenta is not completely impervious to the passage of harmful substances [7], meaning that it is possible that certain contaminants circulating in the maternal blood reach the fetus through the placenta. Some studies have proposed the use of the human placenta as a biomarker of exposure to different potentially toxic elements [8]. It is an easy tissue to obtain since it is discarded after childbirth and allows one to obtain information about the exposure of the mother and the fetus to possible contaminants [9].

The main objective of this study is to test the possible existence of heavy metal deposits in placental samples from the third trimester of pregnancy in women who gave birth in our setting, as well as their relationship with the dietary habits of pregnant women during the final months of pregnancy, and with the perinatal results obtained in the study sample. Our aim was to perform a cross-sectional assessment of the exposure of pregnant women to pollution and contaminants by analyzing the possible deposits of heavy metals in the placental tissue. Therefore, our intention was not to conduct a population study, but rather an exploratory preliminary analysis of the situation in our setting.

## 2. Materials and Methods

### 2.1. Study Population

The maternity hospital of the regional hospital of Malaga is a third-level public center belonging to the Andalusian Health Service, where around 5000 births occur annually. It is the largest healthcare center in the province of Malaga and is the hospital center of clinical reference for the rest of the maternities. The department of obstetrics and gynecology is accredited by the Health Quality Agency of Andalusia and serves women of low and high risk. For detecting the existence of heavy metal deposits in placental tissue, we decided to select a group of 100 pregnant women by consecutive sampling. The participants were selected as they came to give birth in the hospital. The possibility of participating in the study was offered to all pregnant women of legal age with a sufficient level of knowledge of the Spanish language to understand the informed consent offered to them. Only those pregnant women who did not understand Spanish were excluded, as well as those who did not want to participate. Given the added difficulty in processing the samples posed by SARS-CoV2-infection, and not considering it a situation of significant bias to achieve our objectives, we excluded from the study those participants with SARS-CoV2-positive PCRs at the time of admission.

The study was conducted between March and June 2021 and was conducted in accordance with the Declaration of Helsinki, and the protocol (ecarpmp) was approved on 12 July 2016, by the Reference Research Ethics Committee.

As the patients entered to give birth, the purpose of our study was explained to them and they were given information as well as the informed consent document. All women who agreed to participate were given a small survey that collected sociodemographic data (age, level of education, nationality), obstetric and perinatal variables (gestational age, newborn sex, weight, Apgar test and delivery technique), toxic habits (alcohol, tobacco, excessive consumption of caffeine or other drugs) and information regarding dietary habits, with a food consumption survey that determined the frequency of their consumption of dairy, vegetables and fruits, cereals, eggs, meats, fish, mollusks and seafood. We also asked about the type of water they consumed.

### 2.2. Sample Collection

Once the women gave birth, several samples were collected from each placenta, which included both central fragments, around the umbilical cord, and peripheral ones. The samples were homogeneous, about 50 to 100 g, and covered the entire thickness of the chorion, including the maternal and fetal surfaces. The samples were kept in formaldehyde until processing. Obstetric data related to childbirth and newborn were also collected. Once recruitment and the collection of the samples were completed, the samples were processed.

### 2.3. Pre-Treatment of Samples

To ensure our data were accurate, the samples were analyzed at the Central Research Support Services (SCAI). The received placental samples, preserved in formaldehyde, were chopped and briefly washed with quality 1 deionized water (18.2 MΩ⋅cm) to clean the excess formaldehyde and blood. Subsequently, they were frozen at −80 °C to be able to be freeze-dried, and thus, eliminate the water content of the sample without altering the sample due to heating effects. The freeze-drying process was carried out for 96 h, using a LyoQuest model lyophilizer from Telstar (Bensalem, PA, USA). Once freeze-dried, they were milled for 2 min at 875 cycles/min in a ball mill (Mixer/Mill 8000M model of SPEX SamplePrep, Metuchen, NJ, USA) for homogenization.

The freeze-dried and homogenized samples were digested to carry out the analysis of the metals As, Cd and Pb. For acid digestion, a Milestone Ultrawave (Sorisole, Italy) microwave digester was used, using a rotor of 15 positions, 240 °C of temperature and 40 Bar of pressure. Digestion was carried out in a pressurized atmosphere with inert nitrogen gas to avoid unwanted reactions, avoid boiling of acids and avoid cross-contamination. For the digestion of the samples, approximately 0.250 g of the lyophilized sample was weighed on an analytical balance in the digester tube and 3 mL of HNO_3_ (Suprapure quality) was added. Once digestion was complete, the samples were made up to a final volume of 25 mL with quality 1 deionized water (18.2 MΩ⋅cm).

Elemental standards were added to the samples before microwave digestion to assess spike recovery; to carry out quality control of the digestion, and with no certified placental reference material, a control sample was used in each batch of digestion, to which known amounts of As, Cd and Pb were added, to be analyzed later. In all cases, it was found that the recovery of these elements after the digestion stage was greater than 95%.

### 2.4. Analysis of As, Cd and Pb by ICP-MS

The Nexión 300D inductively coupled plasma mass spectrometer (ICP-MS) was used. The ICP-MS procedure is based on a physical method used for the detection of ions in their M+ state. An inductively coupled plasma system is used to generate such ions. The mass spectrum of this ion source was measured by means of a quadrupole mass spectrometer. For quantitative analysis, a calibration line was performed using an external calibration from ten points—a target and nine monoelemental calibration standards of 1000 ppm—in addition to the addition of an internal in-line standard. The instrumental conditions during the analysis were: nebulizer gas flow: 0.92 L/min; plasma gas flow: 18 L/min; auxiliary gas flow: 1.2 L/min; RF ICP potential: 1600 W. The concentration calculation was performed according to the syngistix 2.5 software (Perkin Elmer, Waltham, MA, USA) using the relevant dilution data. As mentioned above, the recovery of the elements As, Cd and Pb in the control samples exceeded 95%. The limit of quantification was considered the lowest concentration of the analyte that could not only be detected but can be quantified with a 95% reliability. The quantification limits of 36, 22 and 32, microg/Kg were determined from 10 reagent blank measurements, corresponding to As, Cd and Pb, respectively. The instrumental detection limits were: As = 0.054 microg/L, Cd = 0.033 microg/L and Pb = 0.048 microg/L.

### 2.5. Hg Analysis Using the SMS-100 Mercury Analyzer

The MERCURY ANALYZER (DT-CVAAS) SMS100 (by Perkin Elmer) is based on the principle of thermal decomposition, amalgamation and atomic absorption described in EPA Method 7473 (DT-CVAAS) [10]. The SMS 100 uses a decomposition furnace to release mercury vapor followed by a chemical reduction step used in analyzers based on traditional liquid mercury samples. In addition, a self-sampling machine with the capacity for 42 samples, both liquid and solid, was used. The samples (freeze-dried and ground) were weighed directly into a Ni capsule, using an analytical balance with the capacity to measure up to 0.1 mg. For the determination of total Hg, a calibration line was performed using a blank and thirteen calibration points prepared from a pattern of 1000 ppm Hg. The instrumental conditions during the analysis were: sample quantity: 0.03–1 g; drying time: 10 s; drying temperature: 300 °C; decomposition Tª: 800 °C; Catalyst Tª and Amalgam: 600 °C. For quality control, a placenta sample was analyzed in a sequence of 10 samples, to which a known amount of Hg was added every 10 samples, as well as a Certified Reference Material (MRC) of oyster. The recovery exceeded 95% in all cases. The limits of detection and quantification for Hg were 0.15 and 0.5 microg/Kg, respectively.

### 2.6. Statistical Analysis

The data obtained from the questionnaires, together with the data from the patients’ medical histories and the data with the concentrations of heavy metals from each placenta sample, were entered into the SPSS (IBM SPSS Statistics, version 25) statistical program for further analysis.

Normality was proved with the Kolmogoroff–Sminorff test and it was observed that the distribution of heavy metal concentrations did not match a normal distribution. Spearman’s Rho coefficient was used for correlations. A Chi-square test was used to compare qualitative variables. Non-parametric Mann–Whitney and Kruskal–Wallis methods were used to compare means between groups according to the number of categories of each variable. A *p*-value < 0.05 was considered for statistical significance. A multiple regression analysis was used to predict fetal weight, including variables that were significantly associated. The Intro method was used for regression analysis.

## 3. Results

### 3.1. Main Characteristics of the Participants 

For the study of heavy metal concentrations, samples from 103 participants who came to the Maternal and Child Hospital to give birth were collected. The participants had an age profile between 18 and 45 years, with a mean age of 31.5 years (SD = 6.1) and a median of 31 years. 

Mostly, the participants were Spanish, 81% of the cases, and completed secondary education, accounting for 51% of the total. Table 1 shows the main sociodemographic characteristics.

### 3.2. Exposure to Potential Contaminants 

Of the 103 respondents, 34 women smoked before getting pregnant and only 17 acknowledged continuing to smoke during their pregnancy, as shown in Table 2. The number of people exposed to tobacco smoke during pregnancy is similar to the number of women who smoked before pregnancy—a total of 36 respondents. The numbers of those who did and did not consume caffeine during pregnancy are practically equal, with those who did consume caffeine being slightly higher in number (59.2% consume vs. 40.8% do not consume). Regarding the exposure of toxins through alcohol, drugs or possible contaminants, the majority (94 cases) denied having been exposed. Asked if they modified their diets during pregnancy, 58 people answered no versus 45 who did. Mineral water was the most frequently consumed type of water, with 74 women indicating they drank this.

Table 3 shows the frequency of the consumption of food groups susceptible to contamination. Mainly, the participants in this study consume dairy, bread, cereals and rice 2–3 times a day, accounting for 64.1 and 63.1% of cases, respectively. The consumption of vegetables is more distributed, from 2–3 times a day to 2–4 times a week (this range represents 85.4% of total cases). Half of the respondents consumed eggs (50.5%) and chicken (60.2%) between 2 and 4 times a week. The red meat, pork and lamb group is limited to weekly consumption (for 30.1 and 25.2% of cases, respectively) or was not consumed, with both groups accounting for 37.9%. Lean sausages also represent a disparity in consumption (28.2% daily consumption vs. 30.1% almost never). The intake of white fish is slightly higher than that of oily fish, being limited mainly to weekly intake. Finally, seafood intake is divided, essentially, between those who almost never consume it or those that do, but monthly.

The gestational age at the time of delivery was 39.2 (SD = 1.2). In 83.5% of cases, spontaneous delivery occurred, with a similar distribution of male and female newborns. Of the five caesarean sections performed, four were elective due to fetal malposition or maternal conditions (Table 4).

Analyses showed detectable concentrations of As in 14.56%, Cd in 44.6%, Pb in 81.5% and Hg in 100% of cases. Mean values are shown in Table 5.

The Pb and As concentrations correlated significantly, with a Spearman’s Rho of 0.91 and <0.001, as did the concentrations of Hg and Cd (Spearman’s Rho: 0.256, *p* < 0.004). The age of the participants was associated with the concentration of Cd, showing a statistically significant positive correlation (Spearman’s Rho: 0.35, *p* < 0.001). 

Placental As concentration was higher in cases of tap water intake (145 ng/g [SD: 54.0] versus 12.5 ng/g [SD: 50.4]), but these differences were not significantly different. We found a significant negative correlation between monthly milk consumption and As placental concentration. Significant correlations are shown in Table 6.

Regarding dietary habits, we observed a positive correlation between Hg concentrations and fish consumption, both oily fish (Spearman’s Rho 0.23, *p* < 0.018) and white fish (Spearman’s Rho 0.35, *p* < 0.001). Hg concentrations were significantly higher in cases where consumption was at least weekly (Figure 1). The mean concentration of Hg in cases of consumption at least weekly was 46.8 (SD: 37.2) for the blue weight and 43.7 (SD: 34.4) for white fish, compared to 32.0 (SD: 27.0) and 27.7 (SD: 25.3) in cases of lower consumption (U of Mann–Whitney: 1623, *p* < 0.026; Mann U–Whitney: 1611, *p* < 0.005).

We also found a significant positive correlation between vegetable consumption and Hg concentration (Rho Spearman: 0.226, *p* < 0.022). In addition, we found a negative correlation between milk consumption and arsenic placental (Rho Spearman: −0.63, *p* < 0.012). On the other hand, we found a relationship between Hg and Cd concentrations and tobacco use during pregnancy. In the group of smokers, we observed significantly higher concentrations of Cd and Hg when the habit was maintained during pregnancy (41.3 (SD: 19.4) vs. 27.2 (SD: 4.4), *p* = 0.02; 55.3 (SD: 46.3) vs. 25.3 (SD: 16.1), *p* = 0.02, respectively). 

We did not observe any association between heavy metals placental and gestational age at the time of delivery. However, we observed a statistically significant negative correlation (Rho Spearman: −0.151, *p* < 0.05) between Hg concentration and fetal weight (Figure 2).

Fetal sex was shown as a confounding factor of this association. Adjusting for sex, in the female group, Hg concentrations in the placenta predicted fetal weight values to a large degree, as seen in Table 7 and Figure 3.

Regarding Pb concentrations, we observed that, in cases where concentrations were above the median value, tap water consumption was significantly more frequent (Chi-Square: 3848, *p* < 0.05) (Figure 4). We found no relationship between these and specific consumptions, toxic habits or prematurity or fetal weight.

## 4. Discussion

This work is a cross-sectional observational study on the possible exposure to environmental pollutants during pregnancy and the concentration of heavy metals in the placenta. We evaluated the levels of four heavy metals, arsenic, cadmium, lead and mercury, in a sample of 103 placentas from women who gave birth at the Regional Hospital in Malaga. The placenta is a biological matrix of exposure to different organic and inorganic pollutants [4] and allows one to obtain information about the exposure of the mother and the fetus to possible contaminants. Moreover, the concentration of heavy metals in the placenta correlates with the concentration in the umbilical cord and in different fetal tissues such as the kidney, liver or brain [6].

Our results confirm the deposit of heavy metals in the placenta at term; although, despite being a homogeneous population, the frequency of exposure seems to be variable. The mean concentration of As for all 103 cases was 44.8 ng/g (SD: 276.0). However, we only found detectable concentrations in 14.56% of the samples, although in these cases the concentrations were high. In one case, we found an excessively high concentration (2756.56 ng/g) without being able to detect any type of specific personal or professional exposure. It should be noted that the newborn was diagnosed with complex heart disease, tricuspid valve atresia. In the remaining cases, in which As was detected in the placenta, the mean concentration was 133.4 (SD: 101.8), higher than that described by other authors in previous studies [7]. In a study of neural tube defects in northern China [8], among controls, placenta arsenic concentrations were 11.8 ng/g on average. In another study, in the participants who lived close to a copper smelter in Bulgaria [9], mean arsenic concentrations of 23 ng/g were detected in the placentas of participants living near the smelter, while an average of 7 ng/g arsenic was detected in controls. On the other hand, a study in Chile [11] on the placental concentrations of heavy metals and fetal growth reports concentrations of 170 ng/g in the control group, and 280 in the fetal growth restriction group, still higher than that detected in our sample. Although the concentration of As was higher in cases of tap water intake (145 ng/g [549.0] versus 12.5 ng/g [SD: 50.4]), these differences were not significant. However, the significant negative correlation between monthly milk consumption and As concentration is striking, in which the possible chelating effects of some milk compounds (taurine, vitamin A or vitamin E) influence this [12,13].

The mean concentration of cadmium detected in our study was 17.9 ng/g (SD: 23.8); however, detectable values were only found in 44.6% of cases, with the average concentration being 40.1 (SD: 19.3) in these cases. The concentrations reported in the literature are variable, ranging from 1 to 6 ng/g and 30 ng/g in the placentas of unexposed populations [4,14]. High concentrations of Cd have been associated with factors such as tobacco consumption [15], and more recently, in a Spanish study [16], with the intake of potatoes in certain areas of the country. In Spain, the prevalence of tobacco use in pregnant women is 26.0% [17,18], ranging from 19% to 34% [19,20]. In our study, the proportion of women who smoked before pregnancy was 33%, and 16.5% maintained this habit. In addition, 35% of patients reported being exposed to tobacco smoke during pregnancy. The relationship of [Cd] with the age of the pregnant woman observed in our study had been revealed in previous studies [21] and could be produced by the effects of chronic exposure. 

Significant concentrations of Hg were found in all placental samples, with the mean value being 38.1 ng/g (SD: 30.4). Previous studies report highly variable Hg concentrations depending on the reference population. Mainly in populations with high fish consumption [14,18] or from industrialized urban areas [14] did these concentrations increase, reaching average values of between 87 ng/g and 144 ng/g, respectively, while in populations with lower exposure and low fish consumption the concentrations are lower, around 1.4 ng/g [20]. We could say that our results are indicative of a significant level of exposure, possibly due to the level of fish consumption and smoking habits. With regard to eating habits, according to the data provided by the Spanish Ministry of Agriculture, Weight and Food, fish consumption has increased in recent years, with the average in the area of study being 22.96 Kg of fish per inhabitant per year, almost 10% higher than that of the previous year [22]. 

The relationship between the levels of Hg [23] and other heavy metals [24] and fetal anthropometric variables manifested in previous studies, in which there is evidence of its transplacental passage [24], which identified some molecule transporters of Hg [25,26]. According to our results, it is possible to indicate a certain effect of mercury exposure during pregnancy on fetal growth, and as in other studies carried out in our country, the effect is more striking in the case of newborn girls [27], pointing to the possibility of a certain genetically determined susceptibility.

We found detectable concentrations of Pb in 84 samples of the 103 studied. The mean concentration of Pb in the global population was 70.08 (SD: 65.2), higher than detected in a previous study conducted in our community [28], in which the concentration of Pb was 41.9 (SD: 14.9). The only variable related to the existence of Pb deposits in the placentas studied was tap water intake. As is known, water can be contaminated from the lead existing both in pipes and in plumbing materials used in building and construction [29]. The impact of the accumulation of Pb on public health, especially on child neurodevelopment, increases concerns about the control of daily exposure to contaminated food and exposure to Pb through food, having established European legislative strategies, which aim to reduce exposure to this pollutant [30,31].

In the interpretation of the results, a series of methodological limitations must be considered. One of them is the type of food consumed questionnaire. As it is a questionnaire conducted through a personal interview, some answers may be subject to desirability bias, and some women may feel self-conscious when it comes to answering truthfully. On the other hand, the number of sociodemographic, consumption and exposure variables is limited, and some variables related to the exposure to heavy metals may no longer be included. In our study, there was a low number of caesarean sections since the sample was not random. This is the result of a consecutive sampling on the days on which the participating midwives were on duty. Finally, the number of participants is limited, and they belong to a single hospital region, so the data must be interpreted considering the geographical context in which the study was carried out. Despite this, our work used a larger sample of placentas selected by consecutive random sampling, from the studies carried out in our community, and highlights the need to develop public health strategies to reduce the exposure of the general population, and specifically of pregnant women, to heavy metals. It should be noted that heavy metal deposits are the result of exposure to environmental contaminants so the ideal situation is one in which the deposits do not exist. A certain concentration of heavy metals has not been established as safe in any biological matrix, although for some metals, blood concentrations have been described as the limits from which clinical manifestations appear. There are no prior descriptions of tolerable heavy metal concentrations in the placenta. On the other hand, the impossibility of quantification does not rule out the existence of the contaminant since the detectable limits are usually lower than the quantifiable limits. Therefore, it is possible that the exposure to these contaminants could be higher than described in our work.

## 5. Conclusions

All the samples had detectable concentrations of at least one of the heavy metals that were analyzed. Especially, Hg deposits were found in all cases. The consumption of tap water and fish and tobacco use were associated with higher concentrations of heavy metals in the placenta. Our findings pose the challenge of studying the effects of As, Cd, Pb and Hg on the future health of children, especially in their neurodevelopment, as well as the effectiveness of public health measures to reduce the exposure of pregnant women to these pollutants.

## Figures and Tables

**Figure 1 ijerph-19-14731-f001:**
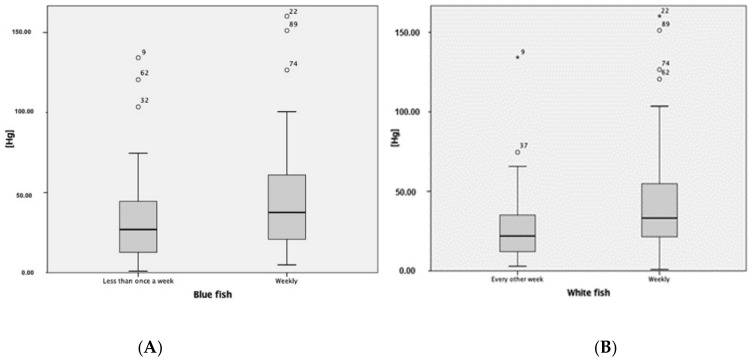
Correlation between Hg concentrations and consumption of blue fish (Spearman’s Rho: 0.23, *p* < 0.018) and white fish (Spearman’s Rho: 0.35, *p* < 0.001). (**A**) The concentrations of Hg detected are significantly higher in those cases in which the consumption of oily fish is weekly. (**B**) The concentrations of Hg detected are significantly higher in those cases in which the consumption of white fish is weekly.

**Figure 2 ijerph-19-14731-f002:**
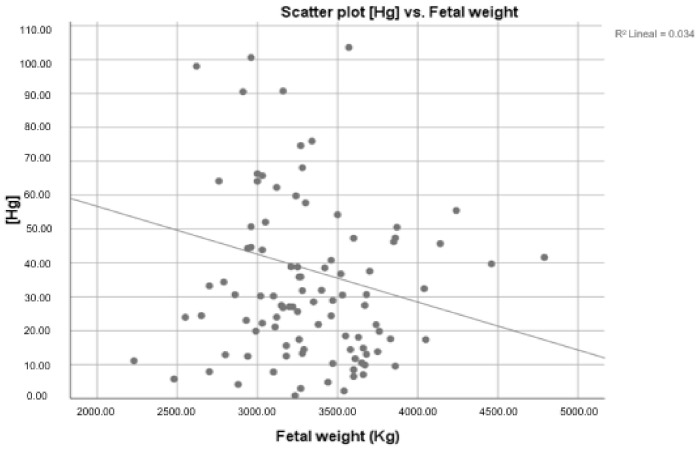
Hg placental concentration according to fetal weight.

**Figure 3 ijerph-19-14731-f003:**
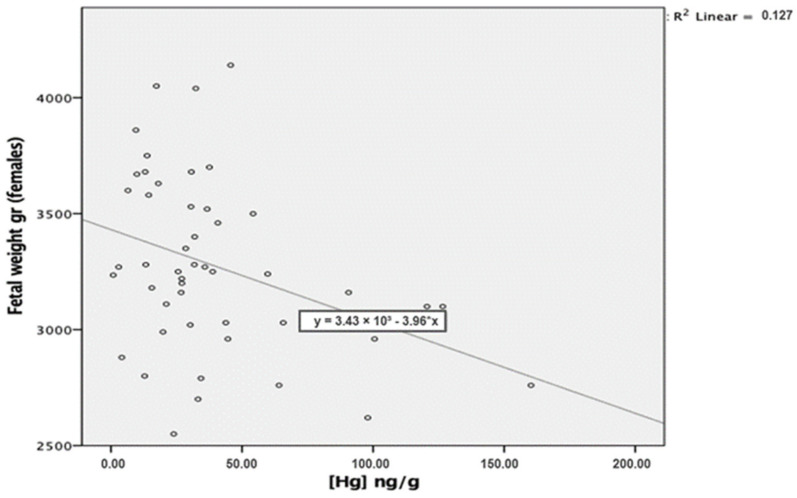
Hg placental concentrations in the female subgroup.

**Figure 4 ijerph-19-14731-f004:**
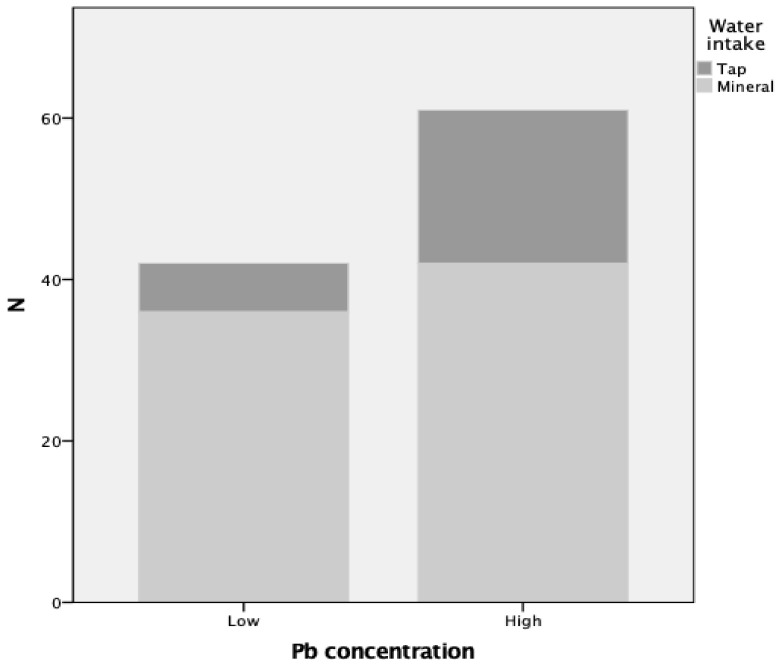
Consumption of tap water and mineral water as a function of Pb concentration.

**Table 1 ijerph-19-14731-t001:** Sociodemographic variables.

Variables	n (%)
Age (years)	
18–24	13 (12.6)
25–29	25 (24.3)
30–34	31 (30.1)
≥35	34 (33.0)
Nationality	
Spain	81 (78.6)
Europe	4 (3.9)
Eastern Europe	5 (5.9)
Asia–Philippines	1 (1.0)
Central–South America	7 (6.8)
Africa–Caribbean	0 (0)
India–Pakistan	1 (1.0)
China	0 (0)
Maghreb–Middle East	4 (3.9)
Education level	
Primary school	26 (25.2)
Secondary school	51 (49.5)
University	26 (25.2)

**Table 2 ijerph-19-14731-t002:** Frequency of possible exposure to environmental pollutants in pregnant women.

Variables	n (%)
Tobacco before pregnancy	
Yes	34 (33.0)
No	69 (67.0)
Tobacco during pregnancy	
Yes	17 (16.5)
No	86 (83.5)
Exposure to tobacco smoke	
Yes	36 (35.0)
No	67 (65.0)
Caffeine during pregnancy	
Yes	61 (59.2)
No	42 (40.8)
Alcohol during pregnancy	
Yes	6 (5.8)
No	97 (94.2)
Substances of abuse	
Yes	0 (0)
No	102 (99.0)
Contact with possible contaminants	
Yes	9 (8.7)
No	94 (91.3)
Working with potential contaminants	
Yes	7 (6.8)
No	96 (93.2)
Type of water consumed	
Tap	15 (14.6)
Mineral	78 (75.7)
Both	10 (9.7)
Modification of the diet during pregnancy	
Yes	45 (43.7)
No	58 (56.3)

**Table 3 ijerph-19-14731-t003:** Frequency of consumption of certain food groups in pregnant women. 1: 4–6 times/day; 2: 2–3 times/day; 3: 1 time/day; 4: 5–6 times/week; 5: 2–4 times/week; 6: 1 time/week; 7: 1–3 times/month; 8: Almost never.

Food Group	1	2	3	4	5	6	7	8
Dairy								
n	4	66	17	6	9	0	0	1
(%)	(3.9)	(64.1)	(16.5)	(5.8)	(8.7)	(0)	(0)	(1.0)
Bread, Cereals and Rice								
n	8	65	22	6	2	0	0	0
(%)	(7.8)	(63.1)	(21.4)	(5.8)	(1.9)	(0)	(0)	(0)
Vegetables (Legumes)								
n	0	1	6	2	46	30	13	5
(%)	(0)	(1.0)	(5.8)	(1.9)	(44.7)	(29.1)	(12.6)	(4.9)
Vegetables								
n	4	26	22	17	23	7	3	1
(%)	(3.9)	(25.2)	(21.4)	(16.5)	(22.3)	(6.8)	(2.9)	(1.0)
Eggs								
n	0	0	5	12	52	25	7	2
(%)	(0)	(0)	(4.9)	(11.7)	(50.5)	(24.3)	(6.8)	(1.9)
Chicken								
n	0	0	6	11	62	14	6	4
(%)	(0)	(0)	(5.8)	(10.7)	(60.2)	(13.6)	(5.8)	(3.9)
Red Meat								
n	0	0	1	2	12	31	18	39
(%)	(0)	(0)	(1.0)	(1.9)	(11.7)	(30.1)	(17.5)	(37.9)
Pork and Lamb								
n	0	1	0	4	18	26	15	39
(%)	(0)	(1.0)	(0)	(3.9)	(17.5)	(25.2)	(14.6)	(37.9)
Lean Sausages								
n	1	5	29	10	16	7	4	31
(%)	(1.0)	(4.9)	(28.2)	(9.7)	(15.5)	(6.8)	(3.9)	(30.1)
White Fish								
n	0	0	0	1	34	32	20	16
(%)	(0)	(0)	(0)	(1.0)	(33.0)	(31.1)	(19.4)	(15.5)
Oily Fish								
n	0	0	0	3	18	22	24	36
(%)	(0)	(0)	(0)	(2.9)	(17.5)	(21.4)	(23.3)	(35.0)
Clams								
n	0	0	0	0	2	10	32	59
(%)	(0)	(0)	(0)	(0)	(1.9)	(9.7)	(31.1)	(57.3)
Prawns								
n	0	0	0	0	3	15	43	42
(%)	(0)	(0)	(0)	(0)	(2.9)	(14.6)	(41.7)	(40.8)
Octopus								
n	0	0	0	0	2	12	39	50
(%)	(0)	(0)	(0)	(0)	(1.9)	(11.7)	(37.9)	(48.5)

**Table 4 ijerph-19-14731-t004:** Obstetric and perinatal variables of the patients.

Variables	N (%)	Variables	Mean (SD)/Mode
Gender		RN Weight (g)	311.8 (423.4)
Female	48 (46.6)	Apgar 1 (min)	9
Male	55 (53.4)	Apgar 5 (min)	10
Delivery			
Caesarean section	5 (4.9)		
Spontaneous	86 (83.5)		
Instrumental	12 (11.7)		
Technique			
Caesarean section	5 (4.9)		
Spatula	3 (2.9)		
Forceps	7 (6.8)		
Vacuum	2 (1.9)		
Type of caesarean section			
Elective	4 (3.9)		
Emergency	1 (1.0)		

**Table 5 ijerph-19-14731-t005:** Concentration of heavy metals measured in microg/Kg in placental samples.

		[As]	[Cd]	[Pb]	[Hg]
Quantifiable (cases)		15	46	84	103
Mean		308.0	40.1	108.1	38.1
Median		92.8	32.3	67.2	30.2
Std Dev		684.4	19.3	225.0	32.4
Percentiles	25	64.6	26.0	42.4	14.9
	50	92.8	32.3	67.3	30.2
	75	189.9	51.0	104.1	47.3

**Table 6 ijerph-19-14731-t006:** Main significant correlations. Significant Spearman’s Rho values (*p* < 0.05).

	[As]	[Cd]	[Hg]
[Pb]	0.91		
Maternal age		0.35	
Milk intake	−0.63		
White fish intake			0.35
Blue fish intake			0.23
Vegetable intake			0.22
Birthweight			−0.15

**Table 7 ijerph-19-14731-t007:** Estimations of the parameters of the simple linear regression model in the concentrations of Hg in the placenta as a function of the fetal weight in the female group.

	Unstandardized Coefficients			
Fetal Weight	B	Std. Error	t	Sig.
(Constant)	624.207	1805.31	0.346	0.000
[Hg] ng/g	−4.072	1.511	−2.694	0.010
G.a. at birth	71.404	45.879	1.556	0.127

## Data Availability

The data presented in this study are available on request from the corresponding author.

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
