# Peer review of "Detection of Relevant Heavy Metal Concentrations in Human Placental Tissue: Relationship between the Concentrations of Hg, As, Pb and Cd and the Diet of the Pregnant Woman"

_ijerph, 2022, doi:10.3390/ijerph192214731_

Round 1

Reviewer 1 Report

The present report examines the presence of 4 potentially toxic heavy metals (Hg, As, Pb, and Cd) in placental tissue in a cohort of pregnant persons delivered at Hospital Regional Universitario de Málaga and relates these to self-reported environmental exposures. The study design and methodology for assessing the heavy metal analytes appears to be adequate, although the consecutive sampling was limited by excluding SARS-CoV-2 positive cases and restriction to the days on which participating midwives were available. Self-report was used to assess potential exposures. The presentation of the results could be improved. The discussion of the study's limitations was well-considered and forthright. Ultimately, this study contributes to a body of knowledge regarding perinatal environmental exposures with public health implications that could help inform related policy decisions. 

Specific Comments:

1. The manuscript would benefit from extensive and careful copyediting to clarify the presentation (e.g.., using editorial software). There are instances where references were made to items that (at least from the materials I was provided) were not presented (e.g., Photo 3 and Photo 4 in lines 114 and 116). There are other instances where the wording is non-standard. For example, when describing the calibration curve for Hg, the authors use the word "white" (line 152) when it is possible the authors intended to use a word with a somewhat different contextual meaning (perhaps "blank"?). This suggestion, however, does not supersede the quality of the scientific work presented, which appears to be solid.  

2. What was the a piori rationale for sampling ~100 subjects meeting the inclusion/exclusion criteria?

3. Given that "detectable" concentrations are a major finding, the authors should report the limits of detection for these analytes. 

4. It was unclear to me how the spike-ins were done. At what point were these added in the processing steps?

5. In Table 1, I am unclear about what " Studies " means (Primary, Side, University) in the table. This does not appear to be explained in the text. Is this a reference to the education level achieved by the participants? 

6. In Table 4, I assume "Sex" (Woman or Man) refers to the fetal sex assigned at birth (male or female). If so, perhaps this could be restated for clarity. 

7. In Table 5, the median is reported, yet in the text (line 216), this is referred to as mean. I am unsure what "Stocking" or "OF" means. The median is reported twice (also in "Percentiles" by definition), and the medians for As differ (92.8 vs. 92.9).

8. A presentation of the Spearman correlations in a table (or tables) would aid the reader in assessing the relationships among the analytes and between the analytes and individual covariates. This could be presented as a heatmap or correlation plot/correlogram.

9. Boxplots might be used to report the data in Table 5 more succinctly. 

10. The Hg data presented individually in Figures 1, 2, and 3 could be consolidated into a single figure. 

11. A table presentation of the regression models would be helpful, as it would provide a systematic overview of the analyses performed. Were any multivariate models used in the statistical analysis?

Reviewer 2 Report

The article presents an interesting  data that cause concern. The authors once again identified the problem of environmental pollution. I would very much like the authors to indicate at least dotted lines ways to solve this problem

On the other hand, despite the detected concentrations of heavy metals, the authors did not reveal any severe pregnancy complications, fetal growth retardation, pretеrm birth and neonatal morbidity in the presented results.

Strengths:

Authors evaluated the levels of four heavy metals such as arsenic, cadmium, lead and mercury in a sample of 103 placentas from women who gave birth.

Limitations:

- No information about the exposure of the mother and the fetus to each possible contaminants, no info about pregnancy course

- There is no information about the newborns and the course of the early neonatal period

-Perhaps it was necessary to discuss in more detail the barrier role of the placenta, since the authors did not identify any associations with adverse outcomes of pregnancy and childbirth

It seems to me that the authors should have indicated the permissible limits of concentrations of four heavy metals and outlined ways to modify the diet in pregnant women

Perhaps the authors will publish a more detailed analysis in the next article

Round 2

Reviewer 1 Report

No further comments other than light editing/proofing.